# Malignant Transformation Rate of Oral Submucous Fibrosis: A Systematic Review and Meta-Analysis

**DOI:** 10.3390/jcm11071793

**Published:** 2022-03-24

**Authors:** Vignesh Murthy, Petros Mylonas, Barbara Carey, Sangeetha Yogarajah, Damian Farnell, Owen Addison, Richard Cook, Michael Escudier, Marcio Diniz-Freitas, Jacobo Limeres, Luis Monteiro, Luis Silva, Jean-Cristophe Fricain, Sylvain Catros, Mathilde Fenelon, Giovanni Lodi, Niccolò Lombardi, Vlaho Brailo, Raj Ariyaratnam, José López-López, Rui Albuquerque

**Affiliations:** 1Department of Oral Medicine, Guy’s & St Thomas’ NHS Foundation Trust, London SE1 9RT, UK; vignesh.murthy@gstt.nhs.uk (V.M.); barbara.carey@gstt.nhs.uk (B.C.); sangeetha.yogarajah@gstt.nhs.uk (S.Y.); richard_james.cook@kcl.ac.uk (R.C.); michael.escudier@kcl.ac.uk (M.E.); 2School of Dentistry, University Dental Hospital, Cardiff University, Cardiff CF14 4XY, UK; mylonasp@cardiff.ac.uk (P.M.); farnelld@cardiff.ac.uk (D.F.); 3Faculty of Dentistry, Oral & Craniofacial Sciences, King’s College London, London SE1 9RT, UK; owen.addison@kcl.ac.uk; 4School of Medicine and Dentistry, University Santiago de Compostela, 15782 Santiago de Compostela, Spain; marcio.diniz@usc.es (M.D.-F.); jacobo.limeres@usc.es (J.L.); 5Oral Medicine, CESPU University, 4585-116 Gandra, Portugal; luis.monteiro@iucs.cespu.pt (L.M.); luism.silva@cespu.pt (L.S.); 6Oral Medicine, University of Bordeaux, 33405 Bordeaux, France; jean-christophe.fricain@inserm.fr (J.-C.F.); sylvain.catros@u-bordeaux.fr (S.C.); mathilde.fenelon@u-bordeaux.fr (M.F.); 7Oral Medicine, Università degli Studi di Milano, 20126 Milano, Italy; giovanni.lodi@unimi.it (G.L.); niccolo.lombardi@unimi.it (N.L.); 8Oral Medicine, School of Dental Medicine, University of Zagreb, 10000 Zagreb, Croatia; brailo@sfzg.hr; 9Oral Medicine, University of Manchester, Manchester M15 6FH, UK; senathirajah.ariyaratnam@manchester.ac.uk; 10Oral Medicine, Faculty of Medicine and Health Sciences, School of Dentistry, University of Barcelona, 08907 Barcelona, Spain; 18575jll@gmail.com

**Keywords:** oral submucous fibrosis, malignant transformation rate, oral potentially malignant disorders, oral cancer, oral squamous cell carcinoma, areca nut, betel nut

## Abstract

Oral submucous fibrosis (OSF) is a chronic progressive condition affecting the oral cavity, oropharynx and upper third of the oesophagus. It is a potentially malignant disorder. The authors collated and analysed the existing literature to establish the overall malignant transformation rate (MTR). A retrospective analysis of medical and dental scientific literature using online indexed databases was conducted for the period 1956 to 2021. The quality of the enrolled studies was assessed by the Newcastle-Ottawa Scale (NOS). A meta-analysis using a random effects model of a single proportion was performed along with statistical tests for heterogeneity. The overall proportion of malignancy across all studies was 0.06 (95% CI, 0.02–0.10), indicating an overall 6% risk of malignant transformation across all studies and cohorts. Sub-group analyses revealed strong differences in proportion of malignancy according to ethnicity/cohort; Chinese = 0.02 (95% CI 0.01–0.02), Taiwanese = 0.06 (95% CI, 0.03–0.10), Indian = 0.08 (95% CI, 0.03–0.14) and Pakistani = 0.27 (95% CI 0.25–0.29). Overall, the MTR was 6%; however, wide heterogeneity of the included studies was noted. Geographic variations in MTR were noted but were not statistically significant. Further studies are required to analyse the difference between cohort groups.

## 1. Introduction

Oral submucous fibrosis (OSF) is a chronic progressive condition affecting the oral cavity, oropharynx and upper third of the oesophagus. It is a potentially malignant disorder, whose malignant potential was first described by Paymaster [1]. It is characterised by epithelial atrophy of the oral mucosa, juxta-epithelial inflammation, and chronic fibrosis of the lamina propria, resulting in the hallmark symptoms of progressive trismus, dysphagia, a burning sensation and intolerance to spicy foods [2,3,4].

OSF appears to have a defined geographical distribution, affectingly populations from predominantly Southeast Asian countries including India, Taiwan, China, Bangladesh, Malaysia, Singapore, Thailand and Sri Lanka, but has also been reported in other countries such as South Africa and Saudi Arabia [2,3,4]. The prevalence of OSF in European countries is low, with cases in European countries predominantly associated with ethnic groups of Asian origin [2,3,4]. In 1996, it was estimated that approximately 2.5 million people had OSF and this figure has doubled to approximately 5 million globally [2,3]. It is considered a public health issue in many Southeast Asian countries [5]. The aetiology of OSF is multifactorial in nature involving environmental factors (capsaicin in chilies, tobacco, and micronutrient deficiencies), genetics and immunological factors. It has been widely established that the main aetiological factor in the development of OSF is the use of areca nut. 

Areca nut is the unhusked whole fruit of the areca nut tree. If the husk is removed and the inner seed or kernel is obtained, this is known as betel nut. The International Agency for Research on Cancer has classed areca nut as a Group I carcinogen [2,6]. Areca nut contains a number of substances such as copper, alkaloids and flavanoid compounds which play a role in producing the histological changes seen in OSF. The alkaloids present in the areca nut are arecoline (main constituent), arecaidine, guvacine and guvacoline. Arecaidine is more potent than arecoline and its availability in the oral environment is increased by the presence of slaked lime [Ca(OH)_2_], which hydrolyses arecoline into arecaidine [2,4]. The flavanoids include tannins and catechins and these have a synergistic effect with alkaloids.

The premalignant nature of OSF was first described in 1956 by Paymaster [1], who observed oral squamous cell carcinomas in a third of a cohort affected by the condition. Pindborg et al. [7] reported a malignant transformation rate (MTR) of 2.8%, whilst another study in 1984 reported a higher MTR of 4.5% over a median follow-up period of 8 years [8]. Murti et al. in 1985 reported an MTR of 7.6% over a 15-year period [1,7,8,9].

The exact risk of malignant change of oral submucous fibrosis (OSF) is unknown and may be compounded by variable figures reported in the literature, which may impact on patient management and prognosis. 

Whilst a recent systematic review and meta-analysis calculated the MTR at 4%, the authors noted the limited availability of studies [10]. “The aim of this systematic review was to collate and analyse the existing literature on oral submucous fibrosis (OSF) and establish the overall malignant transformation rate (MTR). This was achieved by assessing English and non-English articles in the search strategy and adopting the Newcastle-Ottawa Scale (NOS) to assess study quality”. 

## 2. Materials and Methods

The systematic review followed the Preferred Reporting Items for Systematic Reviews and Meta-Analyses (PRISMA) statement guidelines [11]. Key aspects of the protocol are summarised below. 

### 2.1. Protocol Registration 

The current systematic review is registered on the International Prospective Register of Systematic Reviews (PROSPERO), University of York’s Centre for Review and Disseminations under the identification number CRD42021216333. The registered protocol can be accessed at: https://www.crd.york.ac.uk/prospero/display_record.php?RecordID=216333 (accessed on 23 January 2022).

### 2.2. Keywords Selection

A systematic search of the scientific literature was performed for articles published from 1956 to 2021. The literature search was conducted in PubMed, Web of Science, PsycINFO, Google Scholar and the Cochrane Database/Cochrane Central using selected keywords based on the study objectives. A manual search of the literature was additionally undertaken and consisted of a reference list of the selected articles, and a reference list from recent systematic reviews on this topic. After removal of duplicate articles, 90 records in total were retained for screening. The detailed search strategy is available online: https://www.crd.york.ac.uk/prospero/display_record.php?ID=CRD42021216333 (accessed on 23 January 2022) [Mylonas P, Albuquerque R, Murthy V, Yogarajah S, Carey B, Diniz-Freitas M, Monteiro L, Lodi G, Fricain JC. Oral submucous fibrosis: a systematic review and analysis of the reported malignant transformation rates. PROSPERO 2021 CRD42021216333]. Search terms: “oral submucous fibrosis”, plus: “malignant transformation”, “malignancy”, “pre-malignant”, “pre-cancerous”, and “cancer”. Compound logical expressions with Boolean operators were additionally performed on the same fields on the Web of Science: (oral AND submucous AND fibrosis AND transformation AND rate AND (cancer OR malignant OR premalignant OR precancerous)). 

After the extensive literature search, this systematic review utilised both prospective and retrospective cohort studies. No randomised-controlled clinical trials (RCT) were available for use in the systematic review. Papers were selected based upon whether abstracts included information regarding episodes of malignant transformation or association with oral malignancy. 

### 2.3. Studies Selection and Data Extraction Process

An updated database search was conducted by PM and VM. The filtering and screening processes were independently conducted by two investigators (SY and GL) and were supervised by the group leader (RA) who judged and resolved any discrepancies or disagreement. Two review authors (VM and PM) extracted data independently, using a standard data extraction process, and two further authors (RA and GL) cross-checked the resulting database entry against the full manuscript. Reviewer calibration was performed prior to data extraction by asking reviewers to extract data from specific articles and comparing the similarity and consistency of the extracted information. The calibration process involved primary training for the reviewers on the selection and data extraction of studies based on eligibility criteria. Reviewers were asked to judge if a study was eligible or not based on a 20% sample of the studies. Once they had achieved an appropriate level of concordance (inter-examiner agreement of Kappa ≥ 0.81), reviewers independently performed the screening processes. A similar process for data extraction was undertaken, with reviewers calibrating for data extraction on a 20% sample. Once they reached Kappa ≥ 0.81, selection of the relevant data as highlighted on PROSPERO was carried out. 

The inclusion criteria for article selection included all prospective and retrospective cohort studies in any language, with clear details confirming the number of patients subsequently diagnosed with oral cancer. This included case series, meeting abstracts and clinical observational studies. Additional inclusion criteria included the number of patients recruited and patient ethnicity. Individual case reports were excluded from the final systematic review as it is not possible to calculate the relative risk of developing OSF from such reports.

### 2.4. Risk of Bias Assessment

Two review authors (PM and VM) independently assessed and then cross-checked the assessments of risk of bias for each included study (*n* = 16). The NOS was used to assess the quality of the enrolled studies. This was defined by three quality parameters with a total of 9 points [12]. Studies with a NOS score greater than 6 were considered high-quality [13]. Two reviewers (VM and RA) performed the quality assessments separately and, in case of any disagreement, the final decision was resolved by consensus, first among themselves and in case of doubt with the rest of the authors. 

### 2.5. Summary Measures and Methods of Analysis 

All statistical analyses were performed using LibreOffice Calc v3.6 software. The software was used to calculate descriptive measures including mean values for continuous variables. OSF cases were categorised for comparison purposes according to cohort ethnicity. Tests of significance such as the unpaired Student’s *t*-test for comparing means were performed for the raw data sets; a value of *p* < 0.05 was considered statistically significant. 

A meta-analysis using a random effects model of a single proportion was performed following statistical test for heterogeneity. Pooled results of the meta-analysis were performed for each subgroup (based on cohort ethnicity) and for the studies overall. The MTR was based on subgroup and overall group analysis. Confidence intervals were set at 95% and the weighting of each study was calculated as a percentage of all studies included in the meta-analysis. A Funnel Plot was utilised together with Egger’s test to determine study bias and heterogeneity. Additionally, the Trim and Fill method was performed to remove any outlier studies and adjust for any publication bias or missing data sets, and to determine the sensitivity of the original pooled meta-analysis data to any publication bias.

## 3. Results

Of the initial 741 papers identified, including 47 non-English-language studies, a total of 45 papers were selected for full text review (Figure 1). Of these, 29 were subsequently excluded from final systematic analysis. The main reason studies were excluded was that they were case reports discussing isolated rates of transformation. 

Sixteen studies met the inclusion criteria and were included in the analysis [7,8,9,14,15,16,17,18,19,20,21,22,23,24,25,26]. Fourteen out of the 16 studies included were English articles (Table 1). Forty-five non-English-language articles were excluded because they did not meet the inclusion criteria. The vast majority of these 45 studies did not specify one of the following: mean age, gender ratio, criteria for diagnosis and study design. The studies included 16 different patient cohorts. The oldest study used was published in 1964. Seven of the studies were retrospective and nine were prospective. In total, 8516 patients with OSF were evaluated with 780 associated malignancies observed over a variable time period. Gender ratio showed a male predominance for OSF. Six studies were from India, five were from China, five were from Taiwan and one from Pakistan. Quality assessment of included cohort studies based on the NOS is demonstrated in Table 2. 

Using the NOS, the majority of studies scored below 5 (*n* = 13), the lowest scores being 3 (*n* = 2) and three studies scored >6. The reason for low scores was attributed to non-specified follow-up periods and lack of a comparative cohort analysis. The majority of the cross-sectional studies did not have a robust control for differences in their cohort, primarily relating to patient demographics. Many of the studies did not report male to female ratio or mean age to allow for comparison (Table 1). 

Across individual studies, the transformation ranged from 0 to 26.6% with an average of 7.3% as shown in Table 1. The average was calculated by taking the sum of estimated rates of transformation calculated for each study; the number of patients diagnosed with cancer was divided by the respective cohort sizes to give a calculated/estimated transformation rate.

Comparison of the scaling of the reported rates according to cohort sizes showed that the studies carried out on Indian cohorts tended to indicate a higher transformation rate compared with Chinese cohorts.

In order to determine the differences in MTR between differing ethnic cohorts, for each ethnicity we calculated the sum of the number of patients diagnosed with cancer and divided this by the total number of cohorts. This allowed determination of the crude rate of transformation of OSF for each ethnicity observed. Indian OSF cases showed a calculated MTR of 5.4% whilst in China OSF cases showed a calculated MTR of 1.6%, and for Taiwanese cases calculated MTR was 5.6%. A comparison of the differing MTR between Indian and Chinese/Taiwanese cohorts was found not to be statistically significant (two tail, Student’s *t*-test, *p* = 0.232).

The raw cumulative MTR of OSF for all data sets combined, equating to 8516 cases with 780 reported diagnoses of malignancy, was calculated at 9.2%. The total number of confirmed cancer cases was divided by the total number of patients seen in all the included studies, to produce a raw cumulative MTR of OSF using all data sets combined. 

The data considered in the systematic review originated from studies using different cohort sizes. It is possible that larger cohort sizes with longer follow-up periods may lead to a greater confidence in the subsequent recorded transformation rate.

A comparison of the effect of sample size on the reported MTR of OSF by racial group, as seen in Figure 2, was conducted by carrying out both linear and logarithmic regression analysis of the respective number of cases and subsequent proportion of malignancies diagnosed according to ethnicity. Data from Indian cases were considered using logarithmic regression whilst combined Chinese and Taiwanese data were analysed with linear regression. For Indian study data of OSF, there was a strong positive correlation between sample size and percentage of malignancies reported. Figure 3 shows the effects of sample size on the reported MTR; a linear regression analysis was conducted utilising all data sets which showed only a weak negative power trend (R^2^ = 0.0357). 

A random-effects meta-analysis of a single proportion was conducted to determine the proportion of malignant cases within the cohort of diagnosed OSF cases, due to the high level of heterogeneity in the included studies (*I*^2^ = 83%). Results indicated that the overall proportion of malignancy across all studies was 0.06 (95% CI, 0.02–0.10) indicating an overall 6% chance of malignant transformation across all studies and cohorts (Figure 4). Sub-group analysis revealed strong differences in proportion of malignancy according to ethnicity/cohort, Chinese = 0.02 (95% CI 0.01–0.02), Taiwanese = 0.06 (95% CI, 0.03–0.10), Indian = 0.08 (95% CI, 0.03–0.14) and Pakistani = 0.27 (95% CI 0.25–0.29). The funnel plot (Figure 5) was symmetrical and did not indicate presence of high levels of publication bias, although there was a degree of study heterogeneity. Additionally, Egger’s Test (*p* > 0.05) did not suggest the presence of publication bias. The Trim and Fill Method analysis (Trimfill: proportion = 0.09; 95% CI = [0.05; 0.15]) indicated that the overall meta-analysis data did not change significantly and that there little to no effect due to publication bias. 

## 4. Discussion

The results confirm OSF as a potentially malignant disorder; however, the exact rate of MTR could not be accurately calculated. 

There was a male predominance for OSF in all studies, with the exception of a report by Pindborg who showed a female predominance [7]. The gender imbalance can be ascribed, in part, to the higher consumption/chewing of areca nut-based products by males. Hazarey et al. in 2007 [20] conducted a hospital-based cross-sectional study and concluded there was a definite male predominance in OSF cases, with concomitant use of tobacco and areca nut products also being higher in males. This observation is supported by similar studies [1,3,27,29]. Similar to other systematic reviews, there was a lack of information of how the diagnosis was established [10]. 

The geographical variation in the reported rates of malignant change may be attributable to differences in habits associated with areca nut usage. Rates of malignant change were generally higher for cases reported in India versus those reported in China, which is in agreement with the findings of Zhang and Reichart [30]. This may be due to differences in areca nut consumption habits between the two countries and/or the type of areca nut products available. The authors stated that the prevalence of oral cancer is lower in China, compared with India, where no added tobacco products are used in areca nut formulations [30]. Areca nut products vary in their composition between China and India (Table 3). The authors explain that whole areca nut (husks included) is processed by halving the areca nut, and marinating it in different flavours and substances, before it is industrially packaged and sold for general consumption [30]. Areca nut alone (de-husked) is never chewed, instead, whole processed nuts (inclusive of husk) are used for chewing and these are sold in small bags. In China, tobacco is never added to commercially sold areca nut products [30,31,32,33].

In India, there are many different types of freeze-dried commercially available areca nut products, as well as many that are homemade, including paan masala, gutkha, and mawa. Paan masala contains areca nut, betel leaf, calcium hydroxide, and catechu without tobacco. Mawa is a basic mixture of areca nut, tobacco, and lime, whilst gutkha is a combination of dried areca nut, tobacco, and other chemical flavourings [22,29]. Commercially available products have a higher dry weight concentration of areca nut and, therefore, a higher concentration of areca nut per chew. Homemade areca nut products tend to have a lower areca nut content and a reduced concentration of areca nut per chew. Commercial areca nut products are associated with accelerated development of OSF when compared to homemade variants, possibly due to the reduced areca nut content in homemade preparations, together with the use of the betel leaf. Betel leaf contains the antioxidant beta-carotene, which has the capacity to remove free radicals within the areca nut and is considered to have anti-mutagenic properties [22,29].

Yang et al. [28] looked at the chewing habits of individuals in Taiwan using areca nut and concluded that tobacco products are generally not used in conjunction with areca nut products. This was also shown in the study by Zhang and Reichart in 2007 [30] who found fresh areca nut is chewed without tobacco but together with slaked lime and betel influorescene, as well as a betel leaf wrapping. The observation that Chinese and Taiwanese cohorts do not use tobacco products in their areca-nut preparations may explain the lower MTR for these groups in this study. Whilst studies have been conducted looking at the chewing habits of those using areca nut products, a comparison of the effect of concomitant tobacco usage on MTR has yet to be undertaken.

A study conducted by Reddy et al. in 2011 [29] looked specifically at the areca nut chewing habits of a cohort of patients in India, the majority male. Almost half of the cohort evaluated (47%, from Table 2, {22 + 152}/390) reported concomitant use of tobacco with either mawa and or gutkha areca nut products. Combined tobacco and areca nut usage led to the development of more severe forms of OSF compared to areca nut usage alone [29].

Both frequency and duration of areca nut usage seem to play a role in the development of OSF. Reddy et al. [16] suggested that the length of chewing time without spitting during areca nut usage may also influence the onset of development of OSF. A greater severity of OSF was found to be associated with the following: higher frequency and longer duration of areca nut use, and increased duration of saliva retention whilst chewing before spitting. This finding is in agreement with other authors [21]. In contrast, other authors proposed that frequency of use was more important than total duration of usage [30,33,34].

Zhang and Reichart in 2007 [17] stated that the prevalence of oral cancer is lower in China, compared with India. This difference in MTR between different ethnic groups was also confirmed by Zhang et al. in 2012 [35] when they looked at the chewing habits of different Chinese provinces and found the MTR of OSF in Hunan province was lower than in Indian cohorts observed in other studies; 1.2–2% compared with 7.6–13%, respectively. 

In this systematic review, it was found that Indian and Pakistani cases of OSF showed higher MTR compared with Chinese and Taiwanese cases. Additionally, the calculated MTR, which looked at an overall comparison of all OSF cases grouped by ethnicity, indicated that MTR was higher for Indian cohorts versus Chinese and Taiwanese cohorts. This may be due to differences in chewing habits and the higher concentration of areca nut products available, as well as frequent concomitant usage of tobacco in India compared with China and Taiwan. Caution must be exercised when interpreting the results from Pakistan, given only 1 study (of large cohort size) was included in the present analysis. Standardised reporting of research into OSF is required, particularly in regions such as Pakistan, to allow accurate and robust statistical analysis and recording of clinical outcomes [23].

Areca nut products are increasingly being used by children. Gupta et al. in 2013 [36] reported two paediatric cases of OSF in children aged 5 and 12. It is important that more research is conducted to study the effects of paediatric areca nut usage to assess for any difference in MTR compared with adults.

Overall, MTR may be affected by a combination of several factors including composition of the areca nut product, frequency and duration of areca nut consumption, concomitant risk factors for malignant transformation such as tobacco smoking or alcohol consumption, and genetic variation amongst different ethnicities. This study has demonstrated an overall 6% chance of malignant transformation across all studies and cohorts. This is significantly higher than the reported MTR for other oral potentially malignant disorders such as oral lichen planus (1.4%) and oral lichenoid lesions (3.8%) [37]. 

The main limitation of this review is the high heterogeneity across the studies, resulting in variable outcomes which had an impact on the overall calculated MTR. The follow-up period for patients was not reported by the majority of the included studies, thus providing little long-term observational data which may further alter the MTR. The NOS highlighted the low-to-moderate quality of the studies within this systematic review, therefore introducing a high risk of bias. Finally, this review has drawn predominantly on articles written in English. This is an important factor when interpreting the results as a total of 14 of 16 included were English-language articles.

## 5. Conclusions

Overall, the MTR was calculated at 6% (proportion 0.06, 95% CI 0.01–0.10) whilst the crude rate of transformation (raw cumulative rate of transformation of OSF for all data sets combined) was calculated at 9% (SD + 0.1%). The high level of heterogeneity impacted on calculated MTR and reinforces the need for standardisation of reporting. Regional/ethnic variations in the likelihood of malignant transformation of OSF was also noted with Chinese = 0.02 (95% CI 0.01–0.02), Taiwanese = 0.06 (95% CI, 0.03–0.10), Indian = 0.08 (95% CI, 0.03–0.14) and Pakistani = 0.27 (95% CI 0.25–0.29). Additional research is required to ensure robustness of statistical analysis given study heterogeneity and lack of standardised reporting in some countries.

## Figures and Tables

**Figure 1 jcm-11-01793-f001:**
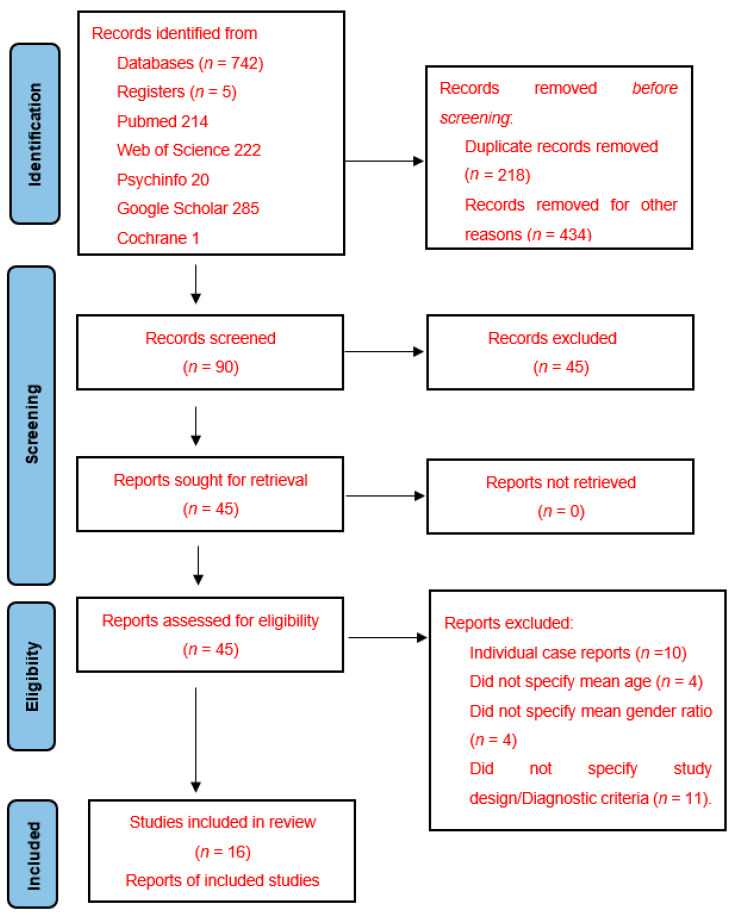
PRISMA flow chart of the screened and included studies.

**Figure 2 jcm-11-01793-f002:**
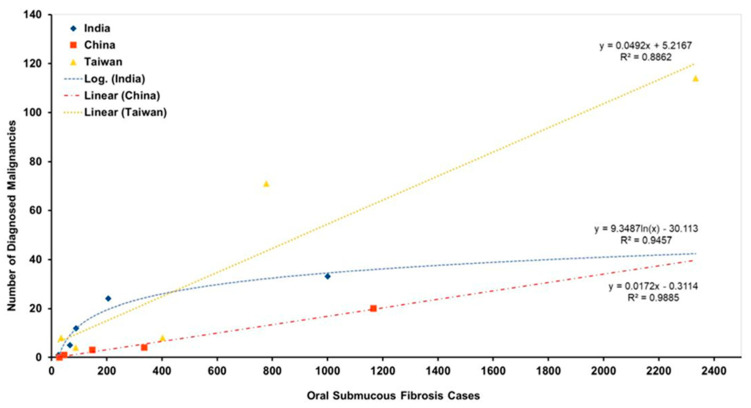
Comparison of cohort size and percentage of malignancies as reported per study. Logarithmic regression utilised for Indian cases and linear regression for Chinese cases.

**Figure 3 jcm-11-01793-f003:**
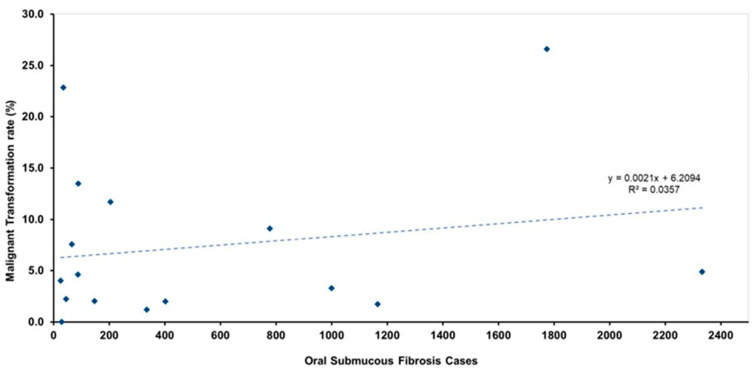
Comparison of cohort size and rate of malignant transformation of OSF as reported per study. Linear regression analysis for all data sets.

**Figure 4 jcm-11-01793-f004:**
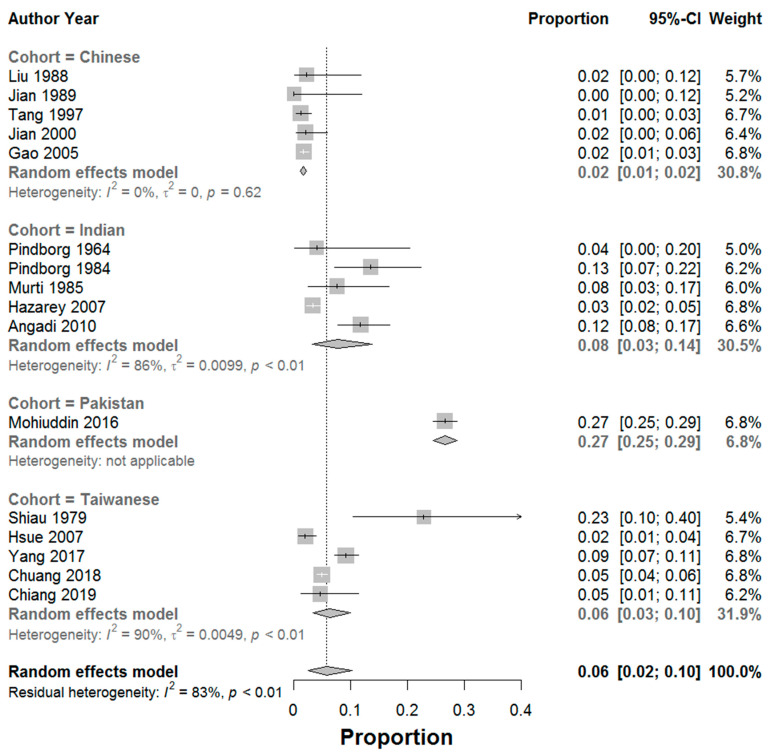
Forrest plot indicating the proportion of malignant transformation of OSF cases (95% confidence intervals) with weighting (%) attributed to each included case. Meta-analysis was performed on a subgroup basis (according to cohort ethnicity) and on an overall basis (including all data sets) [5,7,8,9,14,15,16,17,18,19,20,21,22,23,24,25,26,27,28].

**Figure 5 jcm-11-01793-f005:**
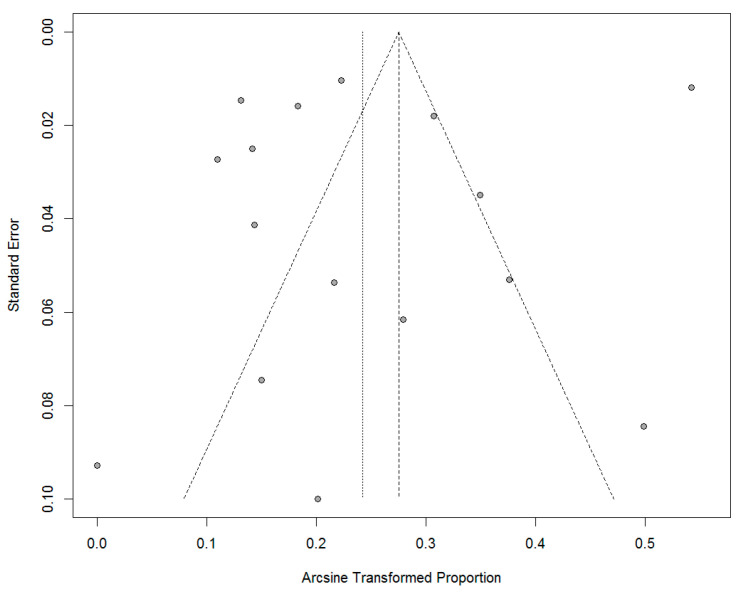
Funnel plot of the standards errors on the *y*-axis plotted against the arcsine of the pro-portion on the *x*-axis. This indicates a broadly symmetrical distribution of studies which suggests a low level of publication bias—there is some study heterogeneity.

**Table 1 jcm-11-01793-t001:** Cohort and patient details of the reviewed articles. NR, Not Reported. Pros, Prospective. Ret, Retrospective. F:M, female: male. MTR, Malignant Transformation Rate.

Study, Year of Study	Cohort Ethnicity [Study Location]	Number of Patients	Number of Patients Diagnosed with Cancer	Mean Age	F:M	Follow Up Period [Years]	Calculated MTR	Reported MTR	Calculated Annual MTR (%) [Reported Annual MTR (%)]	Type of Study
Pindborg et al. [7], 1964	Indian [India]	25	1	41.7	1.5:1	NR	4.0	2.8	NR [NR]	Pros
Pindborg et al. [8], 1984	Indian [India]	89	12	NR	NR	8	13.5	4.5	1.7 [0.6]	Ret
Murti et al. [9], 1985	Indian [India]	66	5	NR	NR	15	7.6	4.5	0.5 [0.3]	Ret
Shiau & Kwan [14], 1979	Taiwanese [Taiwan]	35	8	40.5	1:34	NR	22.9	23.0	NR [NR]	Ret
Liu et al. [15], 1988	Chinese [China]	45	1	NR	1:1.5	NR	2.2	2.2	NR [NR]	Pros
Jian et al. [16], 1989	Chinese [China]	29	0	40.2	1:2.5	NR	0.0	0.0	NR [NR]	Pros
Tang et al. [17], 1997	Chinese [China]	335	4	38.6	1:3	NR	1.2	1.2	NR [NR]	Pros
Jian et al. [18], 2000	Chinese [China]	147	3	NR	1:5.7	NR	2.0	2.0	NR [NR]	Pros
Gao et al. [19], 2005	Chinese [China]	1166	20	37.6	1:5.4	NR	1.7	1.7	NR [NR]	Pros
Hazarey et al. [20], 2007	Indian [India]	1000	33	NR	NR	NR	3.3	NR	NR [NR]	Pros
Hsue et al. [21], 2007	Taiwanese [Taiwan]	402	8	47.5	NR	10	2.0	1.9	0.2 [0.2]	Ret
Angadi & Rekha [22], 2011	Indian [India]	205	24	46	1:11	NR	11.7	11.7	NR [NR]	Ret
Mohiuddin et al. [23], 2016	Pakistan [Pakistan]	1774	472	NR	3:1	NR	26.6	26.6	NR [NR]	Ret
Yang et al. [24], 2017	Taiwanese [Taiwan]	778	71	41.8	1:6.7	6	9.1	9.1	1.5 [NR]	Ret
Chuang et al. [25], 2018	Taiwanese [Taiwan]	2333	114	45	NR	5.7	4.9	0.9	0.9 [0.9]	Pros
Chiang et al. [26], 2020	Taiwanese [Taiwan]	87	4	NR	NR	6.7	4.6	4.6	0.7 [NR]	Pros

**Table 2 jcm-11-01793-t002:** Quality assessment of included studies based on the Newcastle–Ottawa Scale (NOS) for assessing the quality of cohort studies.

Study/Year	Selection (Score)				Comparability (Score)	Exposure (Score)			Total Score
	Representatives of the Expo Sed Cohort	Selection of the Non-Exposed Cohort	Ascertainment of Exposure	Outcome of Interest Was not Present at Start of Study	Based on the Design or Analysis	Assessment of Outcome	Follow-Up Long Enough for Outcomes to Occur	Adequacy of Follow-UP of Cohorts	
Pindborg et al. [7], 1964	1	0	0	1	0	1	0	0	3
Pindborg et al. [8], 1984	1	0	0	1	0	1	1	0	4
Murti et al. [9], 1985	1	0	0	1	0	1	1	0	4
Shiau & Kwan [14], 1979	1	0	0	1	0	1	0	0	3
Liu et al. [15], 1988	1	0	1	1	1	1	0	0	5
Jian et al. [16], 1989	1	0	1	1	1	1	0	0	5
Tang et al. [17], 1997	1	0	1	1	1	1	0	0	5
Jian et al. [18], 2000	1	0	1	1	1	1	0	0	5
Gao et al. [19], 2005	1	0	1	1	1	1	0	0	5
Hazarey et al. [20], 2007	1	0	1	1	1	1	0	0	5
Hsue et al. [21], 2007	1	0	1	1	1	1	1	0	6
Angadi & Rekha [22], 2011	1	0	1	1	1	1	0	0	5
Mohiuddin et al. [23], 2016	1	0	1	1	1	1	0	0	5
Yang et al. [24], 2019	1	0	1	1	1	1	1	0	6
Chuang et al. [25], 2018	1	0	1	1	1	1	1	1	7
Chiang et al. [26], 2020	1	0	1	1	2	1	1	1	8

**Table 3 jcm-11-01793-t003:** Examples of common ingredients most likely to be used together with the areca nut before they are chewed; the ingredients are used in varying quantities [30].

India	China
**Smokeless tobacco**	Betel fruit (husk and leaf)
**Betel leaves**	Peppermint
**Spices**	Gelatine
**Molasses**	Lime
**Catechu**	Calcium carbonate
**Slaked lime**	Calcium hydroxide

## Data Availability

Not applicable.

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
