# Peer review of "Malignant Transformation Rate of Oral Submucous Fibrosis: A Systematic Review and Meta-Analysis"

_jcm, 2022, doi:10.3390/jcm11071793_

Round 1

Reviewer 1 Report

This research included articles that followed a rigorous rule. PRISMA flow chart, PROSPERO registration, critical appraisal, and evaluation of the risk of bias. The meta-regression showed different MTR in different countries. The subgroup analysis calculated the MTR of every country respectively. 

I have a minor suggestion for the authors.
The MTR may affect by many factors like the ingredient of areca nut, gene variants in different ethnicity, and whether they have cigarette smoking or alcohol drinking at the same time.  There is no discussion regarding this part. In addition, I suggest don't calculate the total MTR in a combination with different countries. The acronym in the table has to be explained in the footnote.

Author Response

Thank you very much for your attentive considerations, we hope that the answers are adequate. We attach a file.

José López

Reviewer 2 Report

Thank you for sending me to review this interesting systematic review and meta-analysis evaluating the malignancy of a frequent oral potentially malignant lesion in Asian population. The results of the present study are important observing a malignancy rate superior to other entities of this group such as lichen planus or leukoplakia. Despite this, I would like to comment below on a series of points that should be improved:

Introduction

-Insert the corresponding reference(s) from the sentence lines 58-61.

-Revise the wording of lines 63-65.

-Please in paragraph lines 80-84, insert the corresponding references after each of the annotated studies not at the end.

-The objective of the paper is missing at the end of the introduction.

Methods

-The two links to the PROSPERO pages do not work. Check it in convenience.

-Studies to determine malignant transformation are usually retrospective or prospective studies (longitudinal studies) as they need follow-up. So, I think it is convenient to delete the paragraph lines 119-123.

-Please specify the interexaminer concordance values of the reviewers in the studies selection and data extraction process section.

-Please improve the wording of the inclusion and exclusion criteria. Also include the types of studies included and excluded.

-Publication bias was not analyzed and the analysis include more than 10 studies in the final analysis.

-Please follow PRISMA standards in the flow chart http://prisma-statement.org/prismastatement/flowdiagram.aspx. You should specify:

(1) The items obtained from each database; (2) The reasons for exclusion of studies discarded after full text analysis; among others.

Results

-Review Table 1:

Failure to specify: Number of patients evaluated

Specify if the follow-up time is in months, years....

Specify how patients with OSF were diagnosed in each study.

Provide an explanatory legend for abbreviations or symbols.

-Specify that the heterogeneity is very high >75% in the initial cohort, the Pakistani cohort, and the Taiwanese cohort. As well as in the final meta-analysis.

Discussion

-Review the wording line 294.

-Discuss how the diagnosis of OSF of the cases included in the reviewed studies has been made. I think it is a very relevant point, as it happens in malignancy of other oral potentially malignant disorders for example oral lichen planus or oral leukoplakia.

-Discuss that malignancy rates are higher than those observed in oral lichen planus and leukoplakia.

-I believe that the limitations of the study include the heterogeneity of the studies, which is very high, and the follow-up, which is not reported by most of the included studies. As well as the low/moderate quality of the studies according to the Newcastle-Ottawa scale. Please discuss these limitations.

Author Response

(The authors gave the same response as above.)

Round 2

Reviewer 2 Report

Thank you very much for responding to all suggestions and comments.
They should review before the possible publication the use of acronyms in line 92 page 2. And also reword the objective as the text on lines 93-96 is clearly not an objective but the justification of the objective.
The Prospero page still does not work, I have tried with two browsers "Google Chrome" and "Safari" and in both the answer is "Can not show the document". If it is not in open access it is really useless.

Author Response

Thank you very much for your comments and for reviewing our manuscript, we attach a file in pdf format. And put yourself here in the PROSPERO link if there is any problem when sending it to the file.

https://www.crd.york.ac.uk/prospero/display_record.php?RecordID=216333.

Thanks

José López
